# A Small Set of Nuclear Markers for Reliable Differentiation of the Two Closely Related Oak Species *Quercus Robur* and *Q. Petraea*

**DOI:** 10.3390/plants12030566

**Published:** 2023-01-26

**Authors:** Hilke Schroeder, Birgit Kersten

**Affiliations:** Thuenen Institute of Forest Genetics, 22927 Grosshansdorf, Germany

**Keywords:** *Quercus robur*, *Quercus petraea*, species differentiation, molecular markers, whole genome sequencing

## Abstract

*Quercus robur* and *Q. petraea* are, in addition to *Fagus sylvatica*, the main economically used deciduous tree species in Europe. Identification of these two species is crucial because they differ in their ecological demands. Because of a changing climate, foresters must know more than ever which species will perform better under given environmental conditions. The search for differentiating molecular markers between these two species has already lasted for decades. Until now, differentiation has only been possible in approaches with a combination of several molecular markers and a subsequent statistical analysis to calculate the probability of being one or the other species. Here, we used MiSeq Illumina data from pools of *Q. robur* and *Q. petraea* specimens and identified nuclear SNPs and small InDels versus the *Q. robur* reference genome. Selected sequence variants with 100% allele frequency difference between the two pools were further validated in an extended set of *Q. robur* and *Q. petraea* specimens, and then the number of markers was deliberately reduced to the smallest possible set for species differentiation. A combination of six markers from four nuclear regions is enough to identify *Q. robur*, *Q. petraea* or hybrids between these two species quite well and represents a marker set that is cost-efficient and useable in every laboratory.

## 1. Introduction

*Quercus robur* L. and *Q. petraea* (Matt.) Liebl. are the two predominant oak species in Central Europe. Together, both species make up about 10% of the stands, which makes oaks the second most common deciduous tree species in Europe after beech [1,2]. Among tree species native to Europe, oaks have the highest species diversity of associated organisms at all trophic levels. More than a thousand animal species (including insects, birds, small mammals) live on and with oaks [3]. From an economic point of view, oak wood has a special importance due to its high strength and resistance. It is used as construction timber, as well as processed into barrels, railroad sleepers, furniture, parquet and veneers. During climate change, oaks are thought to be a tree species with high adaptive capacities [4]. Thus, its proportion in European forests may even increase during the next decades [5].

Genetic differentiation of the species *Q. robur* and *Q. petraea* is a difficult task due to their very close relationship and high hybridisation tendency e.g., [6,7]. The search for molecular markers to differentiate these two species has already lasted for decades. First isozymes/allozymes were used for species differentiation back in the nineties until the beginning of the 2000s, e.g., [8,9]. From then on, microsatellites were the markers of choice to observe differences between the two species [10,11]. However, the differentiation only worked on an allele frequency level with low probability. In a dedicated study, isozymes, AFLPs, SCARs, microsatellites and SNPs were used by Scotti-Saintagne [12] to find differences between the two species. They found 389 markers, most of which resulted in low differentiation of the species. A comparison of complete chloroplast genomes of different *Quercus* species revealed that the sequence divergence is low within the genus [13], and even when using a combination of chloroplast and nuclear SNPs, species differentiation is only possible using a high number of SNPs [14]. Nevertheless, four regions were found to be potentially useful as DNA barcodes for species differentiation but were not further applied for the development of easy-to-use markers. Moreover, it has to be considered that hybrid identification is not possible using chloroplast DNA only. All these studies have in common that only a small fraction of the analysed hundreds of markers have discriminating power.

For foresters it is crucial to know whether to plant one or the other species in given environmental conditions because of the different ecological demands of the two species [1]. Thus, the easy-to-use methods for species and hybrid differentiation using genetic markers based on DNA polymorphisms in the nuclear genome, as presented here, will support tree nurseries and foresters.

## 2. Results

### 2.1. Identification of SNPs and InDels for Potential Species Differentiation and Marker Development

Pool-seq data from *Q. petraea* generated by Illumina MiSeq (29X coverage) in this study were compared to Pool-seq data from *Q. robur* (19X) from a previous study [15] to find potential species-discriminating SNPs and InDels. Variant analysis in a mapping of the *Q. petraea* data against the *Q. robur* reference genome [16] resulted in the identification of 4678 SNVs, MNVs and InDels with an alternative allele frequency of 100%. When applying stronger filtering methods (Chapter 4.2), 2651 variants remained, of which about 420 showed no mapping coverage in a parallel mapping of the *Q. robur* Pool-seq data to the *Q. robur* reference genome. A further 750 variants were excluded because they did not show a 100% frequency of the reference allele in the *Q. robur* pool. Based on the analysis of primers (Chapter 4.5), 75 variants remained, showing an allele frequency difference of the alternative allele of 100% between the *Q. petraea* and the *Q. robur* pool.

For marker development based on the remaining variants, 38 primer combinations were selected. Twenty-six of these primer combinations passed BlastN analysis for specificity (Chapter 4.5). In the next step, the derived markers were initially validated using eight *Quercus* specimens per species. For ten markers, this first test was successful and a further validation by genotyping of up to 50 individuals per species was performed. Primer sequences and characteristics of the remaining four primer combinations are presented in Table 1 and the related DNA variants analysed with this primer combinations are presented in Table 2.

The four primer combinations (Table 1) allowed analysis of the six DNA sequence variants (Table 2) that showed 100% frequency difference of the alternative allele between the *Q. petraea* and *Q. robur* pool. The genomic positions of these variants (originally identified based on mapping of the Pool-seq data to the *Q. robur* genome assembly v1) were transferred to a recent version of the *Q. robur* genome assembly (PM1N; [17]) by BlastN analyses of flanking sequence stretches as described in the Materials and Methods. In the case of two variants (QP_miSeq38 SNP1/2), BlastN analysis of the flanking sequences did not provide a suitable hit to PM1N. Thus, the original scaffold positions for these variants, according to the v1-genome assembly, are included in Table 2.

According to an annotation of the recent *Q. robur* genome assembly PM1N, the related variants in Table 2 are predicted to be located in intergenic regions. However, BlastN analysis versus representative *Quercus* genomes at NCBI provided *Q. lobata* gene hits for all variants. The variants in Table 2 are potentially located in introns of the genes with the exception of the two SNPs at Chr9, which are located in the CDS of a putative detoxification 56-like gene (*Q. lobata* gene LOC115972447; XM_031092737.1). These two SNPs result, theoretically, in an amino acid exchange in the predicted protein in the *Q. petraea* individuals compared to the *Q. robur* individuals analysed by Pool-seq (exchange of alanine in *Q. robur* to threonine in *Q. petraea* at amino acid position 125). BlastP of the protein sequence of *Q. lobata* detoxification 56-like (XM_031092737.1) at TAIR11 (https://www.arabidopsis.org/, accessed on 6 May 2022) provided the *Arabidopsis thaliana* protein RHC1 (RESISTANT TO HIGH CO_2_; AT4G22790.1) as the best hit based on the total score (63% identity). This protein is annotated to encode a plasma-membrane-localized MATE type transporter that is involved in CO_2_ signalling during stomatal aperture regulation in *A. thaliana*.

### 2.2. Marker Validation in Extended Sets of Individuals

For the differentiation of the European white oak species *Q. robur* and *Q. petraea*, the four abovementioned regions from the nuclear genome (primers in Table 1, DNA variants and derived markers in Table 2) were further used. In principal, all markers can be analysed by sequencing the related PCR amplicons. Alternatively, for QP_miSeq32 an application of either a polyacrylamide gel or a Genetic Analyzer is also possible because it is a length variation based on an InDel, and QP_miSeq14a and QP_miSeq36 can also be used as PCR-RFLPs (CAPS markers) because restriction enzymes are available for these SNPs (Table 1).

All markers were validated by the genotyping of 39 or 45 specimens of *Q. robur* and *Q. petraea*, respectively, and QP_miSeq32 was validated with an additional 38 *Q. robur* and 32 *Q. petraea* individuals (Appendix A). In the validation, SNP1 of QP_miSeq14a always showed a T, and SNP 2 of the same fragment always had a C in *Q. robur*, whereas *Q. petraea* was the other way around or heterozygous at both sites (C/T), respectively. Alternatively, a restriction enzyme can be used for this marker (Table 3). *Q. robur* always showed an amplicon length of 188 bp using the fragment QP_miSeq32, and *Q. petraea* showed amplicons of 193 bp length or 188 bp length for samples with an origin in the Caucasus region. Additionally, both species can be heterozygous. For QP_miSeq36, the T in *Q. robur* is within the recognition site of a restriction enzyme. This recognition site is abolished in *Q. petraea* where a C is at the related SNP position instead of a T. Moreover, both species can be heterozygous in rare cases. In the fragment QP_miSeq38, SNP_1 has a G in *Q. robur*, whereas *Q. petraea* has an A or both can be heterozygous (A/G) occasionally. SNP_2 of the fragment QP_miSeq38 showed an A in *Q. robur* and a C in *Q. petraea* or, again, both can be heterozygous in rare cases (Table 3).

### 2.3. Assignment Test

The probability of a correct assignment to either of the two species or a group of potential hybrids was tested using as reference populations 45 *Q. petraea*, 39 *Q. robur* and 20 hybridized specimens with information for at least four (to six) of the markers (Appendix A).

#### 2.3.1. Test Population for Validation

As a first test population, five *Q. petraea* specimens and eight *Q. robur* specimens out of the reference dataset were used for a self-assignment test. These test population included one specimen per species with a clear marker combination regarding Table 3 (highest percentage). The other specimens showed discrepancies in one or two different markers. Out of the 13 used test specimens, one *Q. petraea* was assigned to the hybrid group (Table 4), and one *Q. robur* sample was assigned to hybrids. However, both samples had significant exclusion probabilities for all groups; thus, these were false positive assignments. Nevertheless, the assignment of a high percentage to the hybrid group (Table 4) led us to the decision to remove these samples from the reference dataset for further analysis as it was doubtful whether they were pure *Q. petraea*/*Q. robur* (marked in grey in Appendix A).

#### 2.3.2. Assignment of Potential Hybrids

Next, all of the 20 potential reference (first or next generation) hybridized specimens (Appendix A) were used for a self-assignment test. This test population included nine individuals originally identified (morphologically) as *Q. petraea* and 11 morphologically identified as *Q. robur.* All 20 potential hybridized specimens were assigned to the hybrid group (Table 5).

#### 2.3.3. Assignment of Samples with Unknown Species Identity

Finally, an assignment test was performed with 28 individuals with unknown species identity. From these 28 individuals, 10 were assigned to the group *Q. petraea*, 16 to *Q. robur* and two were identified as being hybrids (Table 6).

## 3. Discussion

The differentiation of the species within the genus *Quercus* is a big challenge because of overlapping phenotypes that are due to, *inter alia*, chloroplast capture and hybridisation, though with a low frequency [10,18,19], and, therefore, introgression events when combined with advantageous adaptation [20,21]. When using plastid markers, the haplotypes depend more on the region than on the species; thus, the same haplotypes can be found in the same region in different oak species (especially for *Q. robur* and *Q. petraea*) [22,23]. Therefore, it is not surprising that the classical barcoding markers based on the chloroplast genome have not enough discrimination power for oak species [13]. However, an analysis of nuclear gene loci revealed an even higher mixture of gene pools between the species *Q. robur* and *Q. petraea* compared to distribution patterns of haplotypes for the two species [23]. Using a taxon assignment test with *Q. robur, Q. petraea* and *Q. pubescens*, a minimum set of 26 SNPs with the highest Fst values, or 38 SNPs when randomly chosen, was needed for species identification at a 95% level [24]. Thus, what we experienced when trying to differentiate these species was a complex speciation pattern together with an incomplete lineage sorting [13].

Nevertheless, we successfully selected a small number of nuclear markers out of over 4500 variants, which allowed the discrimination of the species with high probability. Reutimann et al. [24] found no fully fixed allele for any of their investigated species. Our marker QP_miSeq14a, featuring homozygote allelic configuration of the reference allele in all *Q. robur* individuals analysed and homozygote configuration of the alternative allele in most of the *Q. petraea* individuals (others are heterozygote), is also not fully fixed. Nevertheless, the marker showed a discrimination power of 100%. Looking at the other markers, it is obvious that heterozygosity of only one marker reduces the assignment score only slightly (Table 4, e.g., comparing the specimen QUPET_316 with QUPET_267), but with slightly different power of the markers, because a heterozygous QP_miSeq32 reduces the assignment score only from 99.9 to 99.5%, whereas the assignment score decreases to 98.2% when QP_miSeq36 is heterozygous. This can be explained with the percentage of the different alleles given in Table 3. However, a different discrimination power of QP_miSeq36 when comparing the two species (Table 4, specimens QUPET_54 and QUROB_1932) cannot be explained with different percentages because they are identical (Table 3). Thus, one marker being homozygous for the contrary species has more influence in *Q. petraea* than in *Q. robur*. That is comparable with the results of Guichoux et al. [20], who always observed a higher performance when assigning to *Q. robur* than *Q. petraea*, leading to the conclusion that fewer SNPs are needed to identify *Q. robur* than *Q. petraea*. Maybe this can be explained by an asymmetric introgression towards *Q. petraea* resulting in an increase in the diversity in this species rather than in *Q. robur* [20].

With there already being problems identifying the species, identification of hybrids between such two species is an even more difficult task. How to define which is a pure species, when as we showed, both species can be heterozygous in some of the markers or can even show the alleles more frequently for the other species. There are studies using whole genome sequencing to investigate the nature of hybridization where it was shown that hybridization can occur but does not compromise the species integrity, though only small regions of the genome are responsible for the species identity [25,26]. In the *Q. robur*–*Q. petraea* complex, backcrossing is predominant unidirectionally from *Q. petraea,* which is the pollen donor, to hybrids, leading somehow to a regeneration of *Q. petraea* within *Q. robur* [7].

Despite all these described difficulties, with the presented markers, a quite good differentiation of the species and identification of hybrids between them is possible.

## 4. Materials and Methods

### 4.1. Plant Material

For next generation sequencing, we used six individuals of *Q. petraea*, two each from Russia (Caucasus), Ukraine (Crimea) and Germany (one from North Rhine-Westphalia and one from Schleswig-Holstein) (Appendix A).

For marker validation, DNAs of 157 different individuals from the two species *Q. robur* and *Q. petraea* were screened. The individuals were widespread over Europe coming from France, Germany, Finland, Hungary, Ukraine, Belarus and Russia (from the most Western part up to the Ural Mountains). The DNAs were chosen out of over 6000 samples from a reference sample database available at the Thuenen Institute of Forest Genetics (Appendix A). Furthermore, 20 potential hybrid individuals (originally identified as pure species by morphological determination during sampling, identified as hybrids due to admixture values resulting from a STRUCTURE analysis using a set of 453 markers) were chosen from a recent study [14] as a further reference set (Appendix A).

As a test set, cambium or wood samples of 28 individuals with unknown species identity, declared as being either *Q. robur*, *Q. petraea* or *Q. robur* × *Q. petraea*, were used (Appendix A).

### 4.2. Next Generation Sequencing, Read Mapping and Variant Calling

A pool of total DNA from six individuals of *Q. petraea* was sequenced using Illumina MiSeq with 2 × 300 bp paired-end reads with a haploid nuclear genome coverage of 29X (GATC Biotech AG, Konstanz, Germany). Additionally, Illumina MiSeq data from a pool of 20 individuals from 10 locations of *Q. robur* from an earlier study was used [15] (BioProject ID PRJNA269970 at NCBI’s SRA, run accession number SRR3624658). The reads from both pools were trimmed using the “trim reads” tool of CLC Genomics workbench (CLC-GWB; CLC-bio, a Qiagen company, Aarhus, Denmark). Then, trimmed reads of both pools were mapped against scaffolds of the *Q. robur* reference genome assembly V1_2N (https://www.oakgenome.fr/?page_id=587, accessed on 1 July 2022) [16]. Detection of SNV (single nucleotide variants), MNV (multi-nucleotide variants, up to two variants) and InDels was performed in both mappings using the “basic variant detection tool” of CLC-GWB (with a minimal coverage at SNP position of 13 in the *Q. petraea* mapping and of 5 in the *Q. robur* mapping, a length fraction of 0.94 and a similarity fraction of 0.98). For both species, in addition, lower length and similarity fractions (0.9/0.95) as higher coverage thresholds at SNP positions were tried but ended in the first case in a too low specificity and in the second case a too low number of SNPs for further analysis. Variant tables of both pools were merged together with coverage information using *Variant Tools* [15]. The next step was to reduce the dataset to variants with an alternative allele frequency of 100% in the *Q. petraea* pool compared to the *Q. robur* reference genome. Subsequently, the variants with a minimum coverage of 13 in the *Q. petraea* mapping were selected. These variants were further filtered to only select variants showing a reference allele frequency of 100% and a minimal coverage of 5 in the *Q. robur* pool mapping.

### 4.3. Transfer of Variant Positions from Q. Robur Genome Assembly v1 to Q. Robur Genome Assembly PM1N

A total of 50 bp-sequence stretches flanking selected variants were extracted from the reference sequence in the mappings of the *Q. petraea* pool to the *Q. robur* genome assembly v1 [16]. Sequence stretches were analysed by BlastN, using CLC-GWB, versus the annotated *Q. robur* genome assembly PM1N [17]; annotation file https://urgi.versailles.inra.fr/download/oak/Qrob_PM1N_genes_20161004.gff.gz (accessed on 1 July 2022); data downloaded from https://www.oakgenome.fr/?page_id=587 (accessed on 1 July 2022).

### 4.4. Identification of Potential Genes That Include Selected Variants

To identify potential genes that include a selected variant, 2000 bp-sequence stretches flanking the selected variant each in the reference genome assembly PM1N (or in *Q. robur* genome assembly v1, only in the case of QP_miSeq38 SNP1 and QP_miSeq38 SNP2) were extracted and used as query in BlastN analyses at NCBI (https://blast.ncbi.nlm.nih.gov/Blast.cgi?PAGE_TYPE=BlastSearch, accessed on 16 July 2022) using “refseq_representative_genomes” as database (taxid: 3511; *Quercus*). Gene-IDs (“features”) that were assigned to BlastN-hits with 100% query overlap and more than 96% identity and that were assigned to gene models that overlapped the variant position were selected.

### 4.5. Primer Design, DNA Extraction, PCR Conditions, Post-PCR Processing

Primers were designed and then checked for uniqueness using the scaffolds of the *Q. robur* genome ([17]; https://urgi.versailles.inra.fr/blast/?dbgroup=oak (accessed on 16 July 2022), *Q. robur* genome assembly V1_2N). Only primers that specifically mapped to the *Q. robur* reference scaffolds were used for PCR analyses in the subsequent validation procedure. All primer combinations were initially validated with eight individuals each of *Q. robur* and *Q. petraea*. If the variants held on, then a further validation with 50 to 80 individuals per species followed. All finally used primers for species identification are listed in Table 1.

For the DNA extraction, one cm^2^ of a single leaf was ground to powder in liquid nitrogen. Total DNA was extracted, following a modified ATMAB protocol by [27]. PCR reactions for leaf-derived DNA contained ~30 ng template DNA, 10× PCR buffer, 1.75 mM MgCl_2_, 200 μM dNTPs, 0.25 unit DCSPol DNA polymerase (DNA Cloning Service, Hamburg, Germany) and 0.2 μM of each primer in a total volume of 15 μL. PCR was carried out in a Sensoquest Thermocycler (Göttingen, Germany) with a pre-denaturation step at 94 °C for 10 min, followed by 30 cycles of 94 °C for 45 sec, suitable annealing temperature for each primer combination (between 54 °C and 59 °C) for 60 sec, 72 °C for 60 sec and a final elongation at 72 °C for 10 min. PCR amplification products were checked relative to a 100 bp ladder (Life Technologies, Martinsried, Germany) on a 1% agarose gel stained with Roti-Safe GelStain (Carl Roth GmbH & Co. KG, Karlsruhe, Germany). For length polymorphisms based on InDels, genotyping was performed by PCR and amplicon size screening. PCR products were run on an ABI3730 capillary sequencer. Fragment analysis was performed using GeneMarker™ software v. 2.4.0 (Softgenetics, State College, PA, USA).

For SNPs located in recognition sites of restriction enzymes, CAPS (cleaved amplified polymorphic site) markers were derived if suitable. Genotyping using CAPS markers was performed by PCR, amplicon restriction and fragment size screening. After PCR, the restriction digestion reaction contained 10 μL PCR product, 2 μL 10× CutSmart1 buffer, 0.5 μL of the respective enzyme (New England Biolabs, Ipswich, MA) in a final volume of 20 μL. The reaction lasted different times due to the requirement for the respective enzyme. Restriction products were visualized relative to a 50 bp ladder (Life Technologies, Germany, Martinsried) using a 1% agarose gel stained with Roti-Safe.

Genotyping based on all other SNPs was performed by PCR and amplicon sequencing by Sanger (StarSeq GmbH, Mainz, Germany).

### 4.6. Statistical Analysis

The probability of correct assignment of specimens to a reference group was calculated using the computer program GDA_NT 2021 [28]. For this purpose, a Bayesian method following Rannala and Mountain [29] was used. The approach concerns the derivation of probability density of population allele frequencies from the frequencies in samples [30]. The approach assumes an equal probability density of the allele frequencies of each locus in each reference population.

Additionally, the exclusion probability [30] was calculated to avoid a false positive assignment, meaning an assignment to a group though the sample genetically does not comply with the reference data. In cases where the exclusion probability for a sample is below 0.95 (95% confidence interval), a false positive assignment is improbable.

## 5. Conclusions

A set of six markers analysed with four primer combinations was developed to differentiate the two closely related oak species *Q. robur* and *Q. petraea*. Small fragments were chosen to ensure a broad range of application possibilities, e.g., when using DNA extracted from wood or wood products only fragments <200 bp worked well [15]. Thus, here we presented a small set of nuclear DNA markers with high discrimination power for these two closely related oak species that is easy to use in every laboratory at low cost.

## Figures and Tables

**Table 1 plants-12-00566-t001:** Sequences and characteristics of selected primers.

Name	Primer Sequence	Length (bp)	Tm	Genotyping Method	Fragments (bp)
QP_miSeq14a_F	5′ TGT TGA CCA AAA TGG ATA AGA ATT 3′	187	54 °C	Fragment size screening after amplicon restriction with *Mbo*I	QUROB: 187 QUPET: 80/107
QP_miSeq14_R	5′ GTT TGT CTG TCT TGA ATG GCC 3′				
QP_miSeq32_F QP_miSeq32_R	5′ TGA GGG GAA ATC ACA ATT ATG TC 3′ 5′ TGA TGT TCT GTT CTG ATG AAT GAC 3′	193	59 °C	Amplicon size screening (Genetic Analyzer)	QUROB: 188 QUPET: 193
QP_miSeq36_F QP_miSeq36_R	5′ TCA CTT GTT CTA TTT GCA ACA TAT 3′ 5′ TAT TCT GTG TCT GAG TAG GTG ATA C 3	169	54 °C	Fragment size screening after amplicon restriction with *Mse*I	QUROB: 86/83 QUPET: 169
QP_miSeq38_F QP_miSeq38_R	5′ GTA AAT GGT AAT TGA AAA GGC AT 3′ 5′ CCT GAA ACT CTT GTT CAG AAG AT 3′	193	55 °C	Sanger sequencing	

**Table 2 plants-12-00566-t002:** Selected nuclear DNA sequence variants with 100% frequency difference of the alternative allele between the *Q. petraea* and *Q. robur* pool.

Reference	Reference Position	Type of Variant	Ref. Allele	Alt. Allele	Potential Gene Including the Variant (Identification Based on NCBI-BlastN)	ID of Derived Marker
Chr9	45479689	SNP	T	C	LOC115972447; *Q. lobata* protein detoxification 56-like; XM_031092737.1 (variant in exon)	**QP_miSeq14a SNP1**
Chr9	45479691	SNP	C	T		**QP_miSeq14a SNP2**
Chr7	38644432	InDel	-	GCTTC	LOC115974824; *Q. lobata* protein RRC1 isoform X1/X2; XP_030951210.1/ XP_030951211.1 (variant in intron)	**QP_miSeq32 InDel**
Chr2	31588494	SNP	A	G	LOC115974879; *Q. lobata* two-component response regulator ARR12-like; XP_030951285.1 (variant in intron)	**QP_miSeq36 SNP**
scaffold492	52257	SNP	G	A	LOC115974869; *Q. lobata* transcription initiation factor TFIID subunit 5; XM_031095409.1 (variant in intron)	**QP_miSeq38 SNP1**
scaffold492	52287	SNP	A	C		**QP_miSeq38 SNP2**

Chromosomal positions refer to the recent version of the *Q. robur* reference genome assembly PM1N [17] and scaffold positions to the *Q. robur* genome assembly v1 [16]. Reference positions and potential genes including the variant were identified by BlastN as described in Material and Methods. Primers to amplify the variant-including region and methods for genotyping the variant are described in detail in Table 1. The two SNPs in scaffold492 (v1-assembly) are potentially located at Chr2 in PM1N because a BlastN analysis of larger sequence stretches flanking the SNPs (5000 bp each) versus PM1N provided best hits to Chr2. Ref., reference; Alt., alternative.

**Table 3 plants-12-00566-t003:** Results from genotyping of a larger set of DNAs from the reference sample database (SDB) available at the Thuenen Institute of Forest Genetics using the genetic markers from Table 2 analysed and primer combinations from Table 1.

	QP_mi Seq14a SNP1	F (%)	QP_mi Seq14a SNP2	F (%)	QP_mi Seq32 InDel	F (%)	QP_miSeq36 SNP **	F (%)	QP_miSeq38 SNP1	F (%)	QP_mi Seq38 SNP2	F (%)
Reference samples *Q. robur* * (N = 39/77)	TT	100	CC	100	188/188 188/193	96 4	TT CT CC	85 13 2	GG GA	92 8	AA AC	92 8
Reference samples *Q. petraea* * (N = 45/77)	CC CT	84 16	TT TC	84 16	193/193 188/188 188/193	64 18 18	CC CT TT	81 15 4	AA GA	93 7	CC AC	93 7

F, frequency of the presented allele combination in the set of analysed individuals; *, number of validated individuals (for QP_miSeq32 after the backslash); **, primer sequences (Table 1) amplifying this marker region bind reverse complement to the *Q. robur* reference genome PM1N. Thus, alleles shown for this marker are presented in reverse complement configuration compared to the related SNP in PM1N presented in Table 2.

**Table 4 plants-12-00566-t004:** Result of the self-assignment test with the individuals selected from the references of *Q. robur* and *Q. petraea*. The false assigned specimens are given in grey. The markers in the last column are only given with the number not with the full marker name (Table 3). hetero = heterozygous, homo = homozygous, QP = *Q. petraea*, QR = *Q. robur*.

	Scores			Exclusion Probability			Assigned	Divergent
Sample Name	QUPET	QUROB	Hybrid	QUPET	QUROB	Hybrid	to Group	Markers
QUPET_316_1	1	0	0	0.432	1	1	QUPET	All QUPET
QUPET_318_1	0.999	0	0.001	0.581	1	1	QUPET	36 hetero
QUPET_267_1	0.938	0	0.062	0.875	1	0.968	QUPET	14 hetero 32 hetero
QUPET_89_1	0.921	0	0.079	0.945	1	0.993	QUPET	32 hetero, 36 homo QR
QUPET_54_1	0.284	0	0.716	0.989	1	0.960	Hybrid	14 hetero 36 homo QR
QUROB_282_1	0	0.997	0.003	1	0.626	0.598	QUROB	All QUROB
QUROB_293_1	0	0.983	0.017	1	0.718	0.573	QUROB	36 hetero
QUROB_1932_1	0	0.915	0.085	1	0.933	0.876	QUROB	36 homo QP
QUROB_1606_1	0	0.983	0.017	1	0.711	0.571	QUROB	36 hetero
QUROB_1769_1	0	0.963	0.037	1	0.642	0.528	QUROB	14 missing, all QUROB
QUROB_1763_1	0	0.830	0.170	1	0.910	0.472	QUROB	Both 38 hetero
QUROB_1663_1	0	0.830	0.170	1	0.915	0.476	QUROB	Both 38 hetero
QUROB_1658_1	0.001	0.004	0.995	1	1	0.961	Hybrid	32 hetero, 38 homo QP

**Table 5 plants-12-00566-t005:** Result of a self-assignment test with the 20 hybrids also used as reference population.

	Scores			Exclusion	Probability		Assigned
Sample Name	QUPET	QUROB	Hybrid	QUPET	QUROB	Hybrid	to Group
QUPET_697_1	0	0	1	1	1	0.031	Hybrid
QUPET_741_1	0	0	1	1	1	0.007	Hybrid
QUPET_834_1	0	0	1	1	1	0.007	Hybrid
QUPET_1189_1	0.085	0	0.915	1	1	0.667	Hybrid
QUPET_1197_1	0	0	1	1	1	0.012	Hybrid
QUPET_1291_1	0	0	1	1	1	0.010	Hybrid
QUPET_1348_1	0.007	0	0.993	1	1	0.712	Hybrid
QUPET_1527_1	0	0	1	1	1	0.580	Hybrid
QUPET_25_1	0	0.115	0.885	1	0.995	0.749	Hybrid
QUROB_2415_1	0	0.013	0.987	1	0.995	0.141	Hybrid
QUROB_2615_1	0	0	1	1	1	0.008	Hybrid
QUROB_3540_1	0	0.013	0.987	1	0.997	0.152	Hybrid
QUROB_3542_1	0	0.026	0.974	1	0.997	0.528	Hybrid
QUROB_3810_1	0	0.114	0.886	1	0.984	0.322	Hybrid
QUROB_3816_1	0	0.114	0.886	1	0.994	0.325	Hybrid
QUROB_3965_1	0.003	0	0.997	0.999	1	0.005	Hybrid
QUROB_3967_1	0	0.001	0.999	1	1	0.268	Hybrid
QUROB_3976_1	0	0.005	0.995	1	0.998	0.480	Hybrid
QUROB_4101_1	0	0	0.999	1	1	0.956	Hybrid
QUROB_4104_1	0	0.013	0.987	1	0.996	0.142	Hybrid

**Table 6 plants-12-00566-t006:** Result of assignment test of individuals with unknown species identity.

	Scores			Exclusion Probability			Assigned
Sample Name	QUPET	QUROB	Hybrid	QUPET	QUROB	Hybrid	to Group
FG_259_1	0	0.995	0.005	1	0.634	0.471	*Q. robur*
FG_259_2	0.999	0	0.001	0.520	1	1	*Q. petraea*
FG_259_3	1	0	0	0.363	1	1	*Q. petraea*
FG_259_4	0	0.971	0.029	1	0.762	0.355	*Q. robur*
FG_259_6	0	0.809	0.191	1	0.910	0.438	*Q. robur*
FG_259_7	0	0.971	0.029	1	0.756	0.389	*Q. robur*
FG_259_8	0	0.995	0.005	1	0.704	0.482	*Q. robur*
FG_259_9	0	0.809	0.191	1	0.927	0.392	*Q. robur*
FG_259_10	1	0	0	0.372	1	1	*Q. petraea*
FG_259_11	0.001	0	0.999	1	1	0.104	Hybrid
FG_259_12	0	0.975	0.025	1	0.812	0.599	*Q. robur*
FG_259_13	0	0.995	0.005	1	0.604	0.457	*Q. robur*
FG_259_14	0	0.995	0.005	1	0.654	0.436	*Q. robur*
FG_259_15	0	0.975	0.025	1	0.837	0.528	*Q. robur*
FG_259_16	0.999	0	0.001	0.542	1	1	*Q. petraea*
FG_259_18	0	0.995	0.005	1	0.667	0.532	*Q. robur*
FG_259_19	0	0.995	0.005	1	0.546	0.426	*Q. robur*
FG_259_20	0.001	0	0.999	1	1	0.100	Hybrid
FG_259_21	0	0.953	0.047	1	0.618	0.531	*Q. robur*
FG_259_22	0.999	0	0.001	0.543	1	1	*Q. petraea*
FG_259_23	0.999	0	0.001	0.559	1	1	*Q. petraea*
FG_259_24	0.989	0	0.011	0.649	1	0.993	*Q. petraea*
FG_259_25	0.999	0	0.001	0.513	1	1	*Q. petraea*
FG_259_26	0	0.995	0.005	1	0.523	0.409	*Q. robur*
FG_259_27	0	0.995	0.005	1	0.681	0.480	*Q. robur*
FG_259_28	0	0.809	0.191	1	0.914	0.436	*Q. robur*

## Data Availability

The Pool-seq data of *Q. petraea* are available at NCBI’s SRA, BioProject ID: PRJNA914538.

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
