# Peer review of "A Small Set of Nuclear Markers for Reliable Differentiation of the Two Closely Related Oak Species Quercus Robur and Q. Petraea"

_plants, 2023, doi:10.3390/plants12030566_

Round 1
Reviewer 1 Report
The study by Schroeder and Kersten is devoted to the development of genetic markers for differentiation between closely related Quercus robur and Quercus petraea – economically important tree species with a high hybridization tendency. The problem is old and it has not been solved until now. Whole-genome Illumina sequencing was applied to pools of plants of each species. The pool of Q. robur plants was relatively small (6 individuals only), however, the obtained results were validated on a representative set of 157 plants (approximately equal number of samples of two species). The authors proposed reliable test-systems that gave promising results. In general, the manuscript is well prepared and brings valuable results.
There are several comments that should be addressed:
Major comments:
1) Lines 64-65 – “When applying a stronger coverage filter …” – what is the difference from the filter mentioned in previous lines? In Materials and Methods section, I didn’t find this information also.
2) Section 2.3.3 – Were the samples of unknown species identity further assigned morphologically?
3) Section 4.2 – Why there was such a significant difference in minimal mapping coverage for polymorphism calling for Q. robur and Q. petraea – 13 and 3, respectively? The sequencing coverage was close enough for two species – 29 and 19, respectively. In line 252, the minimal mapping coverage for Q. petraea is already 5 instead of 3.
4) Section 4.2 – Also, why was number 13 chosen for minimal mapping coverage for polymorphism calling for Q. robur? I assume that false-positive polymorphisms can be excluded at a lower rate.
Minor comments:
1) Lines 34, 113, 326 – No need in the full name of Quercus robur.
2) Line 67 – “Around further 750 variants” – it is hard to understand, please, rewrite. However, I’m not a native speaker.
3) Line 75 – “per species each” – “each” seems to be redundant, should be removed.
4) Line 130 – “Q. petraea show a C and isn’t cut by the enzyme.” – it is hard to understand, please, rewrite.
5) Table 4, Table 5, Table 6 – The number of significant digits should be the same for all values (except 1 and 0, they can be kept as is).
6) Line 182 – “But, also when looking at nuclear genes a high mixture …” – it is hard to understand, please, rewrite.
7) Section 4.2, line 243 – There is a need for accession number (or link) of Q. robur reference genome, not only a reference to the article.
8) Section 4.5, line 272 – It should be “Primers were” instead of “Primer were”.
9) Section 4.5 – In my opinion, all primer pairs used in the study should be listed in Supplementary Table.
10) Lines 275-278 – “each eight individuals of Q. robur and Q. petraea” – it seems that “each” should be after “of”. In the next sentence – “If the variants hold on then a further validation with 50 to 80 individuals each per species followed.” – “each” seems to be redundant.
11) Please, check English grammar and spelling. Generally, it is fine and the text is easy to read.
Author Response
Reviewer 1:
Comments and Suggestions for Authors
The study by Schroeder and Kersten is devoted to the development of genetic markers for differentiation between closely related Quercus robur and Quercus petraea – economically important tree species with a high hybridization tendency. The problem is old and it has not been solved until now. Whole-genome Illumina sequencing was applied to pools of plants of each species. The pool of Q. robur plants was relatively small (6 individuals only), however, the obtained results were validated on a representative set of 157 plants (approximately equal number of samples of two species). The authors proposed reliable test-systems that gave promising results. In general, the manuscript is well prepared and brings valuable results.
Answer:
Thank you for the evaluation of our manuscript and the helpful comments.
There are several comments that should be addressed:
Major comments:
1) Lines 64-65 – “When applying a stronger coverage filter …” – what is the difference from the filter mentioned in previous lines? In Materials and Methods section, I didn’t find this information also.
Answer:
Sorry, „stronger coverage filter“ was the wrong term. We changed to “stronger filtering methods” and explained it a bit more in chapter 4.2.
2) Section 2.3.3 – Were the samples of unknown species identity further assigned morphologically?
Answer:
No, because we got these samples as cambium or wood not as leaves. In chapter 4.1 it is now described in more detail.
3) Section 4.2 – Why there was such a significant difference in minimal mapping coverage for polymorphism calling for Q. robur and Q. petraea – 13 and 3, respectively? The sequencing coverage was close enough for two species – 29 and 19, respectively. In line 252, the minimal mapping coverage for Q. petraea is already 5 instead of 3.
Answer:
Coverage of 3 has been corrected to 5. It was 5 for Q. robur. We had to find a compromise between a strong enough filter but not a too low number of resulting SNPs. And yes, you are right the sequencing coverages were not so different, but the sequencing result, nevertheless, were it (uneven coverage distribution of the sequence reads over the reference sequence in case of the Q. robur pool). Thus, we got the best results for a minimal coverage of 5 for Q. robur.
4) Section 4.2 – Also, why was number 13 chosen for minimal mapping coverage for polymorphism calling for Q. robur? I assume that false-positive polymorphisms can be excluded at a lower rate.
Answer:
It was 13 for Q. petraea (5 for Q. robur) because of the above-mentioned compromise. And we decided to use the highest possible minimal coverage. When using a lower coverage for Q. petraea it led only to SNPs that had no coverage or not enough coverage in the Q. robur pool in the next filtering step (as described in 4.2).
Minor comments:
1) Lines 34, 113, 326 – No need in the full name of Quercus robur.
Answer: have been changed
2) Line 67 – “Around further 750 variants” – it is hard to understand, please, rewrite. However, I’m not a native speaker.
Answer: “Around” has been replaced with “about”
3) Line 75 – “per species each” – “each” seems to be redundant, should be removed.
Answer: We removed “each”
4) Line 130 – “Q. petraea show a C and isn’t cut by the enzyme.” – it is hard to understand, please, rewrite.
Answer: Sentence has been rewritten.
5) Table 4, Table 5, Table 6 – The number of significant digits should be the same for all values (except 1 and 0, they can be kept as is).
Answer: has been changed
6) Line 182 – “But, also when looking at nuclear genes a high mixture …” – it is hard to understand, please, rewrite.
Answer: Sentence has been reworded.
7) Section 4.2, line 243 – There is a need for accession number (or link) of Q. robur reference genome, not only a reference to the article.
Answer: included
8) Section 4.5, line 272 – It should be “Primers were” instead of “Primer were”.
Answer: corrected
9) Section 4.5 – In my opinion, all primer pairs used in the study should be listed in Supplementary Table.
Answer:
The primer pairs are listed in Table 1 in the results section. We guess, it is a main result of the work, thus, the results section seems to be the right place. We now refer in 4.5 to Table 1.
10) Lines 275-278 – “each eight individuals of Q. robur and Q. petraea” – it seems that “each” should be after “of”. In the next sentence – “If the variants hold on then a further validation with 50 to 80 individuals each per species followed.” – “each” seems to be redundant.
Answer: corrected
11) Please, check English grammar and spelling. Generally, it is fine and the text is easy to read.
Answer: checked
Reviewer 2 Report
Quercus robur and Q. petraea are both ecologically and commercially important tree species with contrasting ecological requirements but frequently interchanged in forestry practice. Their seeds cannot be safely distinguished, so a reliable identification method is highly important. In the light of this fact, the topic of the study is up-to-date, although it would better fit into Forests or another forestry-oriented journal (but it fits into the scope of Plants as well). The study relies on material covering most of the ranges of both species and well-designed methodology. My only doubts are related to practical applicability of the developed markers, which in two cases depend on amplicon sequencing, which is still a tedious and expensive task, and I wonder whether it can be applied in forest nurseries at a large scale (I mean, the analysis itself will be done in specialized lab, but the nurserymen should order it and pay for it).
A general problem I see with this manuscript is extensive genome sharing among white oaks due to long-lasting introgression. Reference samples classified to either pure species (base on phenotype, as I assume) may in fact contain substantial portion of the other species’ genes, and this may apply also to those focused by this study; this may be the reason for false assignment of samples QUPET_54_1 and QUROB_1658_1 (see Table 4). The other issue is what the authors call ‘hybrids’. Commonly, a hybrid is understood to be F1 – is this the case of their ‘hybrid specimens’? As I understand what is written in section 2.3.2, all ‘hybrids’ were initially morphologically classified as pure species (9 Q.p., 11 Q.r.), and then re-classified as hybrids based on the assignment test. Is it so? Because if yes, then it is a kind of a circular argument.
Otherwise, I have just minor comments, and I recommend publication.
l. 26 both references (Bolte et al., Annighoefer et al.) focus on something else than importance of oaks in European forests, I recommend using the citation of a more general source (e.g., European atlas of forest trees by JRC or Euforgen technical guidelines)
l. 27 species diversity of associated organisms?
l. 115-166 please reformulate
l. 154, 182, 200 ‘However,…’ instead of ‘But, …’
l. 160 there is no orange colour in Table 4
section 2.3.3 what are these samples with unknown identity? How were they morphologically classified?
l. 176 not only hybridisation, see Muir & Schloetterer (2005) Mol. Ecol. 14, 549–561.
l. 178 ‘than on the…’
l. 193 well, talking of full fixation with a sample of 157 individuals in total would be exaggerated, even if in within-species variation was found.
l. 230 determination of the hybrid status of the used individuals deserves a closer description, at least in the supplementary materials. In the section 2.3.2, you write that they were all initially classified as pure species.
l. 311 ? the paper of Peatkau et al. does not deal with exclusion probability
Author Response
Reviewer 2:
Comments and Suggestions for Authors
Quercus robur and Q. petraea are both ecologically and commercially important tree species with contrasting ecological requirements but frequently interchanged in forestry practice. Their seeds cannot be safely distinguished, so a reliable identification method is highly important. In the light of this fact, the topic of the study is up-to-date, although it would better fit into Forests or another forestry-oriented journal (but it fits into the scope of Plants as well). The study relies on material covering most of the ranges of both species and well-designed methodology. My only doubts are related to practical applicability of the developed markers, which in two cases depend on amplicon sequencing, which is still a tedious and expensive task, and I wonder whether it can be applied in forest nurseries at a large scale (I mean, the analysis itself will be done in specialized lab, but the nurserymen should order it and pay for it).
Answer:
Thank you for the evaluation of our manuscript and the helpful comments.
Unfortunately, no restriction sites are available for the SNPs in QP_miSeq_38. Thus, Sanger sequencing is the only option. For QP_miSeq_14, we checked again for restriction sites and included a restriction enzyme for this marker in the manuscript. Furthermore, we replaced “amplicon sequencing” with “Sanger sequencing” in Table 1. Every PCR product is an “amplicon” – that was the reason why we used this term. But, this obviously led to misunderstanding.
-------------
A general problem I see with this manuscript is extensive genome sharing among white oaks due to long-lasting introgression. Reference samples classified to either pure species (base on phenotype, as I assume) may in fact contain substantial portion of the other species’ genes, and this may apply also to those focused by this study; this may be the reason for false assignment of samples QUPET_54_1 and QUROB_1658_1 (see Table 4). The other issue is what the authors call ‘hybrids’. Commonly, a hybrid is understood to be F1 – is this the case of their ‘hybrid specimens’? As I understand what is written in section 2.3.2, all ‘hybrids’ were initially morphologically classified as pure species (9 Q.p., 11 Q.r.), and then re-classified as hybrids based on the assignment test. Is it so? Because if yes, then it is a kind of a circular argument.
Answer:
Of course, we are totally aware of the difficulties to differentiate these two species because of introgression as we referred to in the introduction and clearly stated in the last part of the discussion. And yes, we are pretty sure that this is the reason for the false assignment of the two mentioned samples. The differentiation itself is a difficult task and dealing with hybrids even increases these problems. Nevertheless, we tried these markers with an astonishingly high probability of correct assignment.
A hybrid is the offspring of different plants (taxa) that have cross-pollinated; and a first cross-pollination results in a first-generation hybrid (F1) – we totally agree with you. But, there are cross-pollinations afterwards possible between a F1 hybrid and an individual assigned to a pure species – back-crossings (Kremer & Hipp 2020). Are these not also still “hybrids”? Maybe it would be correct to call them hybridized specimens? In this paper, we mean all stages of hybridisation, not least because we don’t know the hybridisation status of the trees we were using. Thus, we don’t mean always F1 hybrids. We included a remark for definition of “hybrid” meant by us in 2.3.2 and changed there to “hybridized specimens”. Is that okay with you?
Citation from Kremer & Hipp (2020):
In the particular context of the Q. petraea – Q. robur complex, in which backcrossing is preferentially
unidirectional, from Q. petraea to hybrids, Q. petraea (the pollen ‘invader’ species) should be progressively ‘regenerated’ within Q. robur (the ‘resident’ species) stands as a result of recurrent
backcrossing after the initial hybridization. A recent demographic and genetic study of the spatial distribution of pure species and admixed forms conducted over two successive generations confirmed the progressive invasion of Q. robur by Q. petraea as a result of hybridization
It is hard to identify hybrids between the two species morphologically because there are all shapes of in between characteristics or hybrids can even look morphologically more or less like a pure species. In this case, the selected samples belonged to a big sampling campaign within a project where some thousand individuals of both species have been sampled. Therefore, not every individual has re-determined morphologically within the project. We don’t really understand your concern with the circular argument. All (some thousands) individuals mentioned above have been classified as the one or the other species during sampling. Then, within the project, quality checks with a high number of markers led to the assumption that some are not pure species. (We added a short description in chapter 4.1 and included the original measured admixture values from a STRUCTURE analysis performed in Degen et al. 2021a; reference 14). These were selected by us and used for an assignment test.
So, the assignment test in our analysis is a confirmation of the original analysis in this other project (Degen et al. 2021a; reference 14). If you mean that by circular argument, then, yes, it is a kind of. But, how should you define hybrids for references when not by testing them with different molecular and subsequent statistical methods?
In general, “hybridisation” is a most exciting subject and always reason for intensive discussions, but the main issue of our manuscript isn’t hybridisation. In the case of this manuscript, we originally even hesitated if we should include “hybrid” identification. But, the success of our assignment tests led us to the assumption that we can do it. We mitigated the possibility of hybrid identification in the abstract to “quite good” as we already described it in the last part of the discussion.
---------
Otherwise, I have just minor comments, and I recommend publication.
- 26 both references (Bolte et al., Annighoefer et al.) focus on something else than importance of oaks in European forests, I recommend using the citation of a more general source (e.g., European atlas of forest trees by JRC or Euforgen technical guidelines)
Answer: We replaced the two references with the recommended ones.
- 27 species diversity of associated organisms?
Answer: done
- 115-116 please reformulate
Answer: done
- 154, 182, 200 ‘However,…’ instead of ‘But, …’
Answer: replaced
- 160 there is no orange colour in Table 4
Answer: Sorry, it was changed to grey.
section 2.3.3 what are these samples with unknown identity? How were they morphologically classified?
Answer: We got these samples as cambium or wood not as leaves. In chapter 4.1 it is now described in more detail.
- 176 not only hybridisation, see Muir & Schloetterer (2005) Mol. Ecol. 14, 549–561.
Answer: Yes, it is much more. We included at least “chloroplast capture”.
- 178 ‘than on the…’
Answer: changed
- 193 well, talking of full fixation with a sample of 157 individuals in total would be exaggerated, even if in within-species variation was found.
Answer: We are talking about “not” fully fixed alleles, not about full fixation (now line 199).
- 230 determination of the hybrid status of the used individuals deserves a closer description, at least in the supplementary materials. In the section 2.3.2, you write that they were all initially classified as pure species.
Answer:
We already explained a bit above answering your first recommendation. Furthermore, we added a short description in chapter 4.1 and included the original measured admixture values from a STRUCTURE analysis performed in Degen et al. 2021a in Supplementary Table S3.
- 311 ? the paper of Peatkau et al. does not deal with exclusion probability
Answer: Sorry, wrong citation, it is Cornuet et al. 1999 [21] – has been changed